# The Impact of COVID-19 on Hospitality Industry in Greece and Its Treasured Santorini Island

Nikola Medová *[ID], Lucie Macková [ID] and Jaromir Harmacek [ID]

Department of Development and Environmental Studies, Palacký University Olomouc,
779 00 Olomouc, Czech Republic; lucie.mackova@upol.cz (L.M.); jaromir.harmacek@upol.cz (J.H.)
* Correspondence: nikola.medova01@upol.cz

**Abstract:** This paper focuses on the dynamic of the recent upheaval in the tourism and hospitality sector due to the COVID-19 epidemic in Greece and Santorini island. It uses the case study of a country one-fourth of whose GDP consists of tourism. We compare the available statistical data showing the change in variables in the previous years with 2020 and look into the new challenges and opportunities posed by the drop in the numbers of visitors and flights. We focus mainly on the economic and social impact on the destination and possible future scenarios for further development in the area. Data show a significant effect of the pandemic on multiple variables, such as the long-term trend of the importance of tourism sector in GDP in Greece, the number of flights and visitors to Greece and Santorini island, and the contribution of tourism and travel to GDP. Based on the available data, we also construct three foresight scenarios that describe the possible futures for Santorini island in terms of the pandemic evolution. These scenarios may help various stakeholders and policymakers to be better prepared for different developments that may appear.

**Keywords:** tourism; environment; COVID-19; Greece

## 1. Introduction

Greece is a country located in Southeast Europe that is largely sought as a favorite tourist destination in both its variants, in mainland and also its islands. Its GDP consists of three main branches, agriculture, industry, and services. From a long-term point of view, services make up around 70% of total GDP [1]. In many places, Greece faces over-tourism that brings several problematic consequences in economic, social, and environmental regard [2].

One of the most popular Greek islands Santorini is well-known for one of the world's most remarkable active volcanoes. According to Friedrich (2009), there is no other place on earth where geologists can learn so much about volcanism as on this island located in the Aegean Sea. Because Santorini is the only inhabited caldera and due to its stunning architecture, the island entices people from all around the world. From a historical perspective, Santorini had been a peripheral island and had been cut off from the mainland civilization for a long time; therefore, the development of the island went its own way. In the 1970s, the island mainly opened up for tourism and from then it has started changing slowly. All island development has happened due to increasing touristic demand. In recent years, the number of incoming tourists has increased also as a result of social media and therefore created high stress on the environment [3].

Generally, the direct contribution of tourism to the Greek gross domestic product (GDP) typically averages 11.7%, and tourism accounts for 16.7% of employment [1,4]. The country was expected to welcome a record 32 million foreign travelers in 2018—up from just 6.2 million in 1998 and 15 million in 2010. No major European destination has seen a larger increase in visitor numbers this decade, according to the article published in Traveller [5]. Santorini is the most visited Greek island, followed by Crete, Corfu, and Rhodes. The most expensive island, Mykonos, is in the fifth place.

There are two research questions addressed by the paper: first, we want to find out what the impacts of the COVID-19 pandemic on the hospitality industry in Greece and Santorini are. In this context, we would also like to discuss the economic and social impacts the pandemic may have through its effects on tourism. Second, we want to investigate what are the possible scenarios of future development in Santorini.

The purpose of the research is to demonstrate how the COVID-19 pandemic affected the tourism and hospitality industry in Greece and Santorini and also to design possible scenarios of future development in terms of the pandemic for Santorini. This paper analyzes the impact of COVID-19 on tourism and the hospitality industry, which is the main source of income for the local people of Santorini, and importantly, tourism and the travel industry creates around 20% of total GDP in Greece according to statistical data [1]. The next section reviews the literature on the topic of the tourism industry and its impacts. Next, we discuss the impacts of tourism and their possible gaps in events of high uncertainty, such as the current global pandemic. We will demonstrate the impact on economic data, first on Greece and then on Santorini, as an example of a destination extremely dependent on tourism. From the chosen indicators, we analyze the share of travel and tourism (TT) on total GDP and employment and changes that happened due to the pandemic. Additionally, we show the dependence on tourism in Greece through other data such as "spending of international tourists" and "domestic tourism expenditure" and their decline in 2020 compared to previous years. We demonstrate the fall in the number of incoming tourists on both levels, Greece and Santorini. We analyze the data from the year 2020 aiming to show how it might be affecting especially in the economic and social field. Finally, we will discuss the sustainability of the tourism industry and what future scenarios might look like using the foresight method.

Income from tourism is undoubtedly connected to the number of incoming tourists and the quantity of services provided. Regarding the statistical data, Santorini (Thira) is part of NUTS 2, called South Aegean, and NUTS 3, formerly also known under the name Kyklades Prefecture. Since Santorini is there with other nine islands, it is rather difficult to assign the island exact numbers. According to a number of sources [6–8], over 90% of Santorini's total GDP consists of tourism and travel. Therefore, the impact of COVID-19 on Santorini and a connected problem with decreased number of incoming tourists brings a significant problem for islanders from many perspectives.

## 2. Literature Review

There have been many articles and studies on Greek tourism and also on Santorini island. Santorini is famous especially for its geological history, unchangeable architecture, and tourism's impact on the island. Additionally, the myth of the lost Atlantis civilization is connected to this place; therefore, the brand of Santorini is sold more easily. One of the most important sources for this paper has been the statistical data collected from Santorini municipality, Santorini airport, the European Commission, and tourism journals (Greece Is, Traveller, National Geographic). Websites such Santorini365, Greeka, and Santorini-view were used for general information that is provided in the text. Among the academic articles that have been used, the studies by Lichrou et al. (2017), Wadih (2005), or Schagrin (2018) can be named. The book by Walter Friedrich (2009), Santorini: Volcano, Natural History, History, Mythology, presents a deep insight into the topic. Regarding the data provided for Greece, we worked predominantly with Statista and World Travel and Tourism Council data. We have also used current articles covering the tourism trend in Greece and impacts of the COVID-19 pandemic.

### 2.1. Tourist Intentions and Dependency on Travel and Tourism in Greece and Santorini

The intention to travel is influenced by different factors, one of which is the perceived risk of travel [9]. While the risks involved in tourist activities can take different forms, the most widely perceived risk in 2020 was possibly the threat of contracting COVID-19. The travel restrictions put in place by various governments also influenced the propensity

to travel and led to an unprecedented decrease in tourist numbers. Greece and Santorini have long been perceived as popular spots on the Mediterranean tourist map. While there are high effects of the seasonality of tourism and land degradation, the experience of place-making in this destination serves different visions of tourism and development [10]. The local actors play an important role in the process and in what the authors label as the phases of "romancing tourism, disenchantment and reimagining tourism".

Some authors [11,12] discuss the risk of man-made and natural disasters and their long-term influence on tourism. The destination image plays an important role in this model, and some places have a higher risk of natural disasters such as tsunami or earthquakes. The intention to travel might even be influenced by a celebrity endorsement [13]. Overall, since the 1980s, Santorini in particular had been experiencing an enormous growth of tourists coming to the island every year. When the pandemic of COVID-19 emerged in 2020, it obviously hit the island of Santorini and influenced the nearly 70% of local people that depend on the tourism industry. The island has entered into yet another phase in 2020.

### 2.2. Tourism, Environment, and Sustainability in Greece and Santorini

Greece with its rich economic, religious, and intellectual activity for more than three and half millennia has indeed many attractions to offer to tourists. The area is characterized by natural beauty, mild climate, and cultural heritage, which lead to seasonal tourism and attractiveness of the country that without tourism would be a remote and peripheral and poor economic region in Europe. The tourism sector plays a crucial role in economic and regional development [14].

The climate where Santorini belongs creates the best conditions for seasonal tourism, which peaks during the summer months. The history of tourism dates back mainly to the 1960s when the first groups of tourists, usually from the Scandinavian countries, started coming to underdeveloped Santorini. The beginning of the tourism on Santorini was a period of small-scale tourism and was a time when tourists (seen as guests) were received by hosts and experienced personal and authentic moments [10]. As the fame of the place was increasing, Santorini had started developing its infrastructure that could provide convenient conditions for people coming to explore the new place. At the beginning of the 1980s, Santorinian tourism experienced a large boom, and the number of tourists started increasing rapidly. The reason for this growth was the production of an American movie Summer Lovers (1982), which was filmed on Santorini island. In 1972, the military airport turned into a civilian one and started receiving the first civilian flights. Since Santorini had not had any port where larger ships could anchor, the municipality agreed on building the main island port Athinios in the early 1980s. Since then, it has been the main Santorinian port where most of the cruise ships, ferries, and speed boats land.

Athinios port had to be built at the caldera side mainly because of the depth of the sea—it is the only place where the heavy ships could land. However, the location brings certain complications. The space for the hinterland is very limited, and the access road is not maintained and also not safe, since it is a very curvy and steep road. These disadvantages lead to the fact that during the summer season when there is heavy traffic, the road is usually blocked, impassable, or clogged up. Due to this reason, many tourists lose their boat connections because they are not on time to get down to the port.

To simplify the conclusion of the research of the Santorini tourism by Lichrou et al. (2017) [10], basically, we can say that the past is a romanticized period with a warm approach toward tourists, and the tourism performed back then is seen as amateurish, whereas the present is characterized by vanishing traditional ways of life and mass tourism that brings negative aspects into the society and also changes the physiology of the island.

One of the main examples of the close relationship and connection between tourism and environmental degradation is the event from April 2007, when the cruise ship Sea Diamond with over 1200 passengers sank off the coast when it collided with the volcanic rock while trying to anchor in Santorini [15]. According to the newspaper article and interviews with locals, tourists did not bother about the environmental pollution coming

out from the sunken ship. Basically, no response coming from the municipality followed the catastrophe in terms of limiting the number of cruises or tourists coming to the island. The number of cruises anchored in the caldera waters was reaching up to 10 per day just a few years ago. Finally, in 2019, the new regulation appeared and stated that the maximum number of daily cruise arrivals will be maximum 8000 visitors. The mayor said the overflow of tourists has put too much strain on the island's infrastructure and supply. Nearly half of the 2 million visitors coming to Santorini are day visitors who come on cruise ships that stay on the island an average of seven hours. The island attracts more than 10,000 visitors a day on some of the summer's busiest days [16].

*2.3. Over-Tourism and Its Relationship with Santorini and Greece*

We can find multiple definitions for over-tourism based on different variables and tourism indexes. Increasing over-tourism in popular tourist destinations has been a rising problem in recent times. This phenomenon causes many troubles not only in the tourism industry but also in local communities. Usually, it "describes a situation in which the impact of tourism, at certain times and in certain locations, exceeds physical, ecological, social, economic, psychological and political capacity thresholds" [17]. Over-tourism is identified as a multidimensional problem with multiple impacts (environmental, social, and economic). When pointing out a destination in Greece that is experiencing over-tourism, Santorini is the example. Santorini is a well-known Greek island with increasing tourist arrivals every year, especially during recent years, and the boom of social media, and in that sense, we can state that over-tourism could decrease the quality of the tourist product (the island) and therefore cause undesired consequences for all stakeholders linked to the tourism industry. In the case of Santorini, this would mean most business owners on the island.

Santorini tourism had experienced a boom in the last few years. The overall trend in the number of passengers coming to the island was increasing until 2020, a year deeply affected by the COVID-19 pandemic. From approximately 2,300,440 people transported to the island in 2019 (including December 2019), the number dropped rapidly to 569,914 in 2020 (without December 2020) [18].

Generally, tourism is an important industry that makes around 20% of Greece's annual GNP. The number is much higher for Santorini, over 90% whose total GDP consists of tourism and travel. Every decision that the Santorinian municipality issues influences the country's budget. The regulation for the cruise ships caused frustration among the ship operators and other business owners linked to this field [16]. On the other hand, there are many factors influencing the volume of Santorinian tourism. When looking at the statistical data, the numbers of international air arrivals had been continuously increasing from 2010 until 2018 [18]. In 2019, due to restrictions posed by the municipality, the number of one-day visitors from international cruise ships lowered slightly [19,20].

In 2020, the COVID-19 crisis has taken by surprise most of the countries around the world. One of the worst influenced sectors has definitely been tourism. The borders of many countries closed down for a few months, and almost all flights were canceled. In April, normally, Santorini gives evidence of a rise of international flights, and the tourist season usually starts up. In April 2019, there were over 50,000 passengers traveling with international flights, whereas in April 2020, there were zero people coming to Santorini. This 100% drop means a significant reduction of income for every person that is involved in tourism, which practically means at least two-thirds of the population.

Tourism is the fastest growing industry in the world and usually provides many benefits to the destination, such as employment and foreign currency to the host place as well as contributing to the country's GDP. However, tourism often has negative impacts as well, but these are often overlooked in favor of the economic benefits. Among these negative impacts could be named, for instance, environmental degradation, disappearing traditions, and high prices [21].

Tourism's contribution to climate change and its other environmental effects are pressing issues that need to be addressed by implementing policies that would lead to sustainable tourism. However, another way to make tourism sustainable is by reducing the number of tourists. There are two Sustainable Development Goal targets (8.9 and 12.b), which aim at developing and implementing policies and tools to monitor sustainable development impacts for sustainable tourism that creates jobs and promotes local culture and products.

The impact of the COVID-19 epidemic and its restrictions could be described globally. According to the Industry Pulse Report (2020), the estimated impact of COVID-19 was around 100 million job losses and 2.7 trillion USD decline of GDP for 2020. These data were based on middle June 2020 data. Three possible scenarios for the gradual opening of international borders with respect to the level of international arrivals in 2020, relative to 2019 performance, were established as 58% decline in comparison with early June 2019, 70% decline in September, and 78% decline in early December. When we compare these three expected scenarios with the tables based on the real statistical data, we can see that the decline of tourism in Santorini island was as expected [22].

There is a need to question the linear growth model of tourism, as advocated by the World Tourism Organization (UNWTO). The UNWTO is an international organization that is responsible for advancing the SDGs and yet it represents a platform that advocates for growth advocacy [23]. The UNWTO defines sustainable tourism as "tourism that takes full account of its current and future economic, social and environmental impacts, addressing the needs of visitors, the industry, the environment and host communities". Tourism itself should be a genuine driver of solidarity and development [23,24].

## 3. Methodology

The research objective is to analyze the impacts of the COVID-19 pandemic on the tourism industry in Greece and Santorini. We analyze the available data both for Greece and Santorini and demonstrate the impacts of the COVID-19 pandemic using variables such as the number of international and domestic visitors, the contribution of travel and tourism to GDP in Greece (by domestic and international spending), and the spending of international and domestic tourists in Greece. Given the high dependency of Greece and particularly Santorini on tourism, and given the impacts of the COVID-19 pandemic on tourism, we then discuss the possible economic and social effects this may have in Greece and Santorini. Moreover, we also present foresight scenarios of possible future development.

Foresight scenarios are used to enhance our understanding of different future phenomena and can inform the public and other stakeholders and influence their decision making [25]. They can be used to provide different views of the future in different contexts such as this one, when we know certain variables but have to provide narratives about projections of specific types of future. For example, Ratcliffe characterizes scenarios as providing alternative images (rather than extrapolating trends from the present), embracing qualitative perspectives as well as the quantitative data, and evaluating sharp discontinuities [26]. For this paper, we only use three short-term scenarios (usually the most precise) to outline the situation of Santorini in the current year and beyond. Three is the minimal number of different scenarios that are normally used. We use Peter Schwartz's approach to creating scenarios, which includes the driving forces and their ranking by importance and uncertainty [27]. Based upon a combination of unfolding events, we use the data for 2020 and assess the extent of changes that might take place in different directions.

## 4. Findings

The following text is based on the statistical results that provide evidence of Greece and Santorini's dependency on travel and tourism and illustrate the change in basically all travel and tourism data in 2020. The following figures generally show a positive or constant trend until 2020, when a significant change (usually a drop) was recorded. However, we start with Figure 1, which demonstrates the importance of travel and tourism industry by

its contribution to GDP and employment in Greece. Data for these variables were available only until 2019 when the contribution actually peaked at 21.2% of the total GDP and at 26.7% of the total employment.

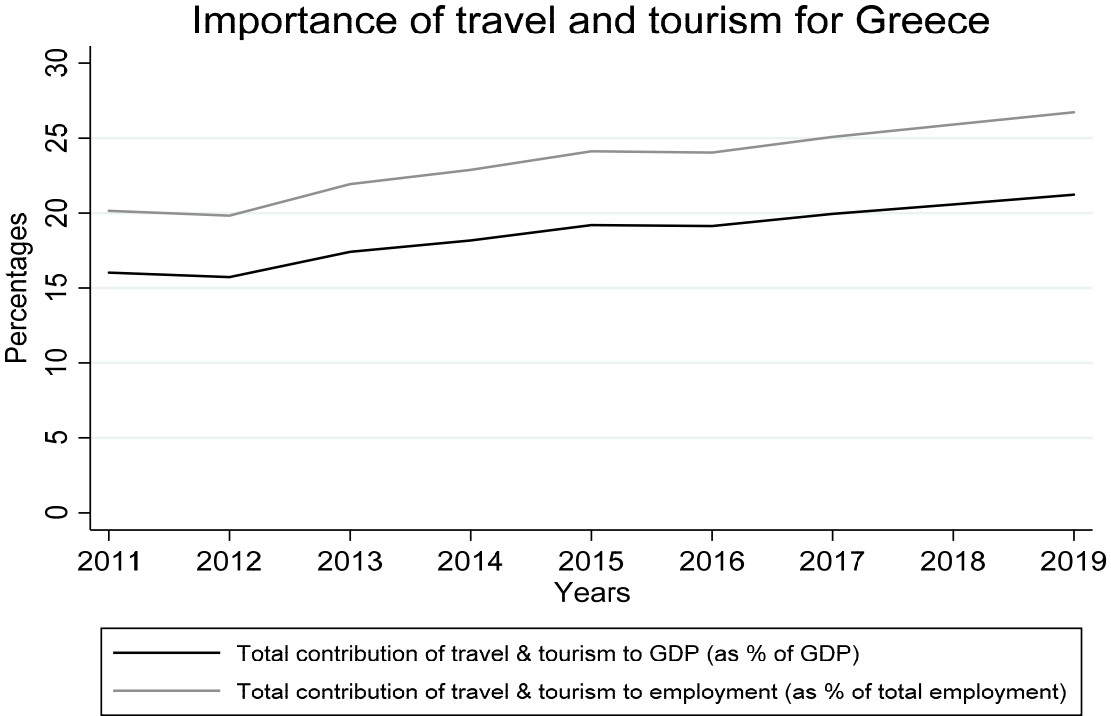

**Figure 1.** Total contribution of travel and tourism to GDP and employment in % (2011–2019). Source: WTTC, 2021, [28].

When looking at the number of international visitors in Greece (Figure 2), the values for the last decade prove that international tourism had been increasing and recorded up to 34 million people in 2019. The drop in 2020 is impressive but not striking when we compare it with the worldwide situation connected to COVID-19 and significant restrictions on international travel. Any other reason for an explanation of such an exceptional drop is hard to find and is rather improbable.

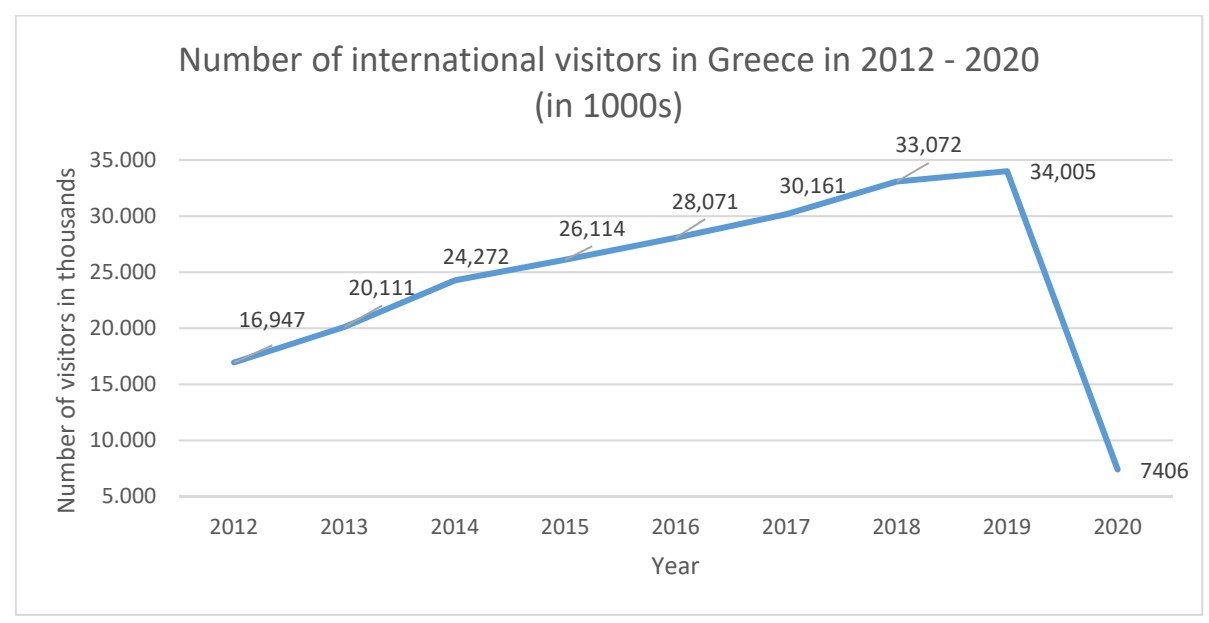

**Figure 2.** Number of international visitors in Greece in 2012–2020 (in 1000s). Source: Statista 2021, [29].

Figure 3 shows the share of domestic and international visitor spending in Greece. As noted, it can be seen that domestic share had been historically lower than the international visitor spending, which strengthens the fact that international tourism plays a more important role in Greek national income. In 2020, the situation changed, and the results turned in another way due to more severe restrictions in international travel and tourism.

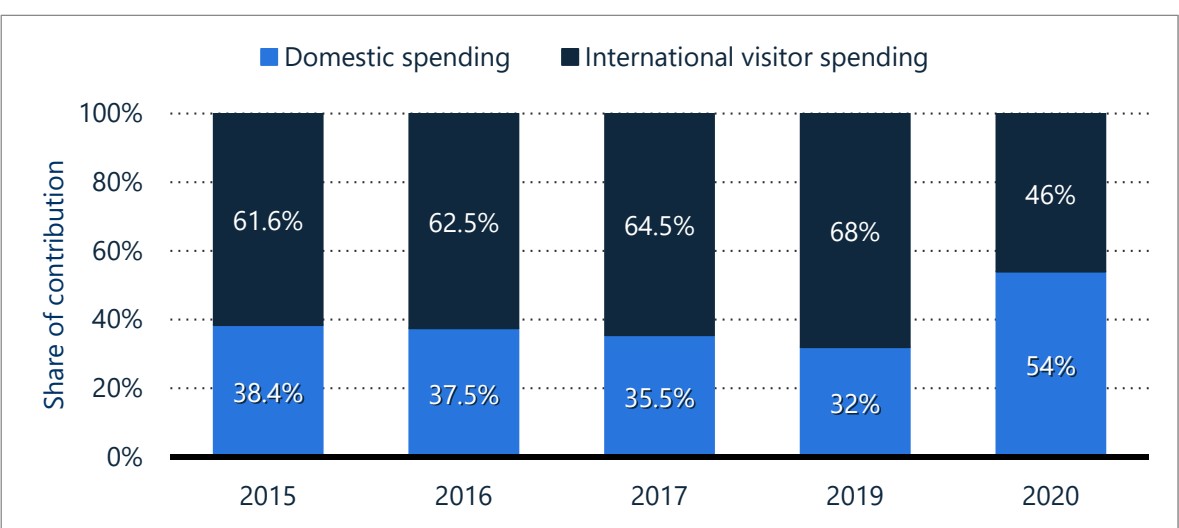

**Figure 3.** Contribution of travel and tourism to GDP in Greece from 2015 to 2020, by domestic and international spending. Source: Statista, 2021, [29].

Figure 4 complements the previous story, and it shows that domestic tourists spent significantly less in absolute numbers over 2012–2019 than international tourists. In 2020, however, both types of expenditures dropped considerably. The decline was much more dramatic for the international expenditures, which went from 23.2 billion euros to only 5.4 billion, while the domestic expenditures decreased from 11.1 to 6.4 billion euros.

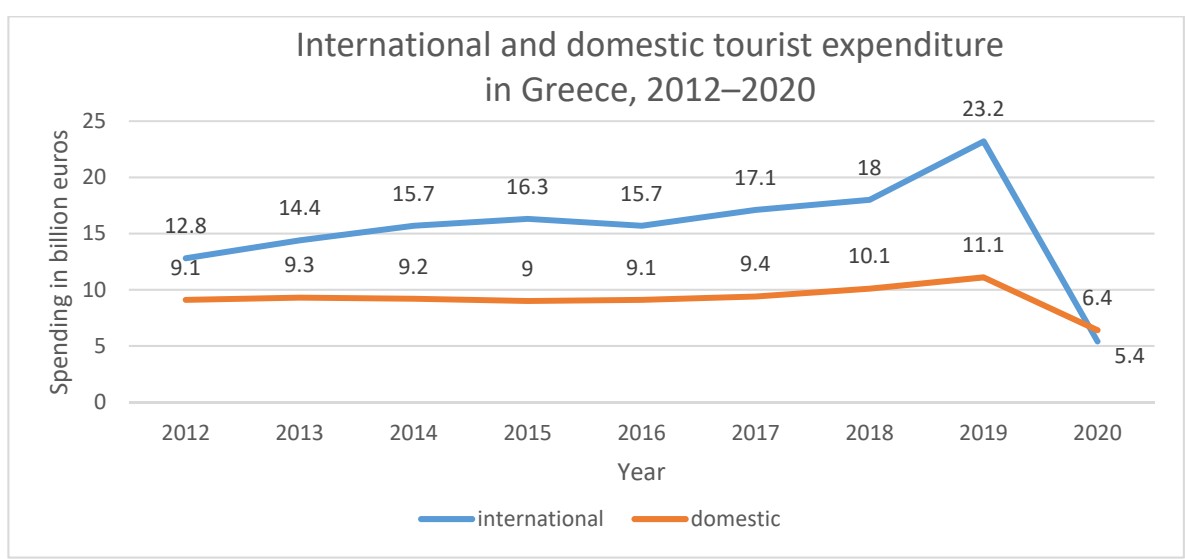

**Figure 4.** Spending of international and domestic tourists in Greece 2012–2020. Source: Statista, 2021, [29].

While previous figures illustrated impacts of COVID-19 on Greece, Figure 5 shows its impact on the hospitality industry in Santorini. Both parts of the figure demonstrate high seasonality in terms of domestic and international flights, and domestic and international passengers to Santorini. It can be clearly confirmed from the data that touristic season starts in April and ends in October. This had been the case until the beginning of 2020.

It can be seen that COVID-19 had three effects. First, the off-season decline in terms of domestic flights and domestic passengers was much deeper in early 2020. Second, COVID-19 extended the off-season drop related to international flights and passengers. Third, the season started later in 2020, its peak was also much lower (half of the preceding years), and the culmination was very quick (there is a spike visible in the charts).

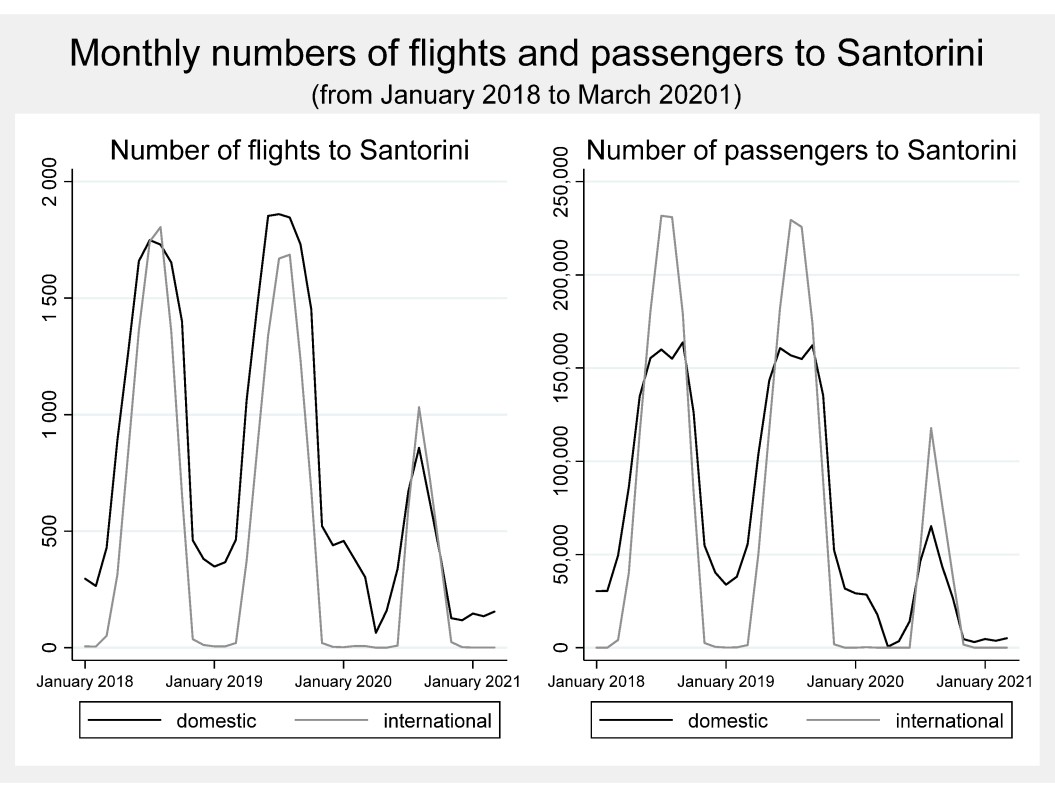

**Figure 5.** Monthly numbers of flights and passengers to Santorini. Source: JTR Santorini Airport, 2020, [18].

## 5. Discussion

### 5.1. Economic and Social Impact

The importance of tourism for the Greek economy is undeniable [1]. There is no doubt that the interconnectedness between sectors and services contributes to the national income. The graphs above suggest that the decline in the tourism sector was clearly caused by the COVID-19 pandemic and there is no reason to believe that the cause should be of a different nature. In this respect, Greece can be considered a country model from the Mediterranean region, which is dependent on tourism.

The economic impact of tourism on Santorini and its locals is crucial. It is also higher in Santorini compared to other Greek islands. As written above, nearly 75% of inhabitants depend on the income from tourism. Based on different SWOT analysis and research, Santorini, unlike other parts of Greece, does not suffer from the lack of job positions and has been a place where many young people find a work placement. Additionally, we cannot state that the economic crisis that has been affecting the whole of Greece since 2009 has played a significant role in Santorini. Since the island has been attractive for people from all around the world, the financial crisis did not hit Santorini itself. On the other hand, the COVID-19 pandemic influenced tourism the most. Therefore, the locals had two options: first, to open up their businesses and hope for as long a season as possible and as many tourists as possible. This, unfortunately, did not happen, since the borders for international passengers opened in June, travel agencies and tour operators had significant financial problems and the COVID-19 positive cases from different countries influenced the restrictions and bans on traveling globally, so the majority of the charter flights had

been cancelled by October. The fixed costs to run a business in Greece are generally high; therefore, there is a need for a long and prosperous season. Second, they could have not opened their businesses (tavernas, restaurants, and hotels) and decided to survive one year from the money saved from the previous years. This might have worked if the future vision of tourism in 2021 was rather optimistic. In December 2020, the situation regarding the upcoming year was still unclear, but with 90% of the GDP of Santorini coming from the tourism industry, the season and opening up for tourists have to come [30,31].

Additionally, the social impact of tourism on the island is inherent. The island used to be a rather poor place until tourism appeared in the late 1960s. When the infrastructure started improving in the late 1970s and at the beginning of the 1980s, the local people changed their focus and left the agricultural and mining life, started to rent their houses, and built new family businesses and hotels. Since then, nearly 75% of inhabitants have depended on income from tourism [30]. That means that local people strongly depend on the income from tourism, and both the volume and length of the season are crucial for surviving the winter times and also new investments or developing current capacities and infrastructure.

By 23 March 2020, restrictions on movement throughout the country were enacted and the official lockdown in Greece started. Over the following few weeks, people wanting to leave their homes were required to send a text message to a government-issued phone number with a code, having only six possible reasons for their movement. Among the acceptable reasons were: travel to or from one's workplace (during work hours), going to the pharmacy or supermarket, visiting a doctor, and personal exercise. "For the most part, everyone followed these measures without complaint. But with the approach of the summer tourist season, compliance quickly turned to worry. More than a quarter of Greece's GDP comes from tourism and the thought of losing out on the much-awaited and much-needed income from the millions of tourists who would arrive in Greece this year was terrifying for some" [31–33].

Unlike other years, Santorini was closed to international flights until the middle of June 2020. Therefore, the season could not start earlier. Normally, hotels open up for the first tourist in May and the season officially finishes at the end of October. Usually a six-month-long touristic season takes place every year and its length and volume are crucial for the local people. In 2020, the season started in July and the capacity of the island had not been fulfilled at all. Many hotels stayed closed, although the exact number is not known. According to JTR Santorini Airport statistics, the overall tourism in 2020 decreased by approximately 65% in comparison with 2019 [18].

As of May 2021, there is no accurate information about the upcoming season and statement of Greece. The EU members talk about so-called COVID passports that will allow tourist travel to some member countries. According to some media channels, Greece will end lockdown measures and open to tourists. Each member state will make its own decision and will allow to travel or recommend not to based on their own experience and also a daily increase in new COVID cases [34,35].

*5.2. Possible Scenarios*

The elements of alternative futures are pulled together into certain number of scenarios. In our case, it is three. The alternative scenarios are most important, and different plausible futures result from uncertainties, including major differences from the present, the value of key quantities, and implications that will lead from each of them. When making statements about the future, the problem of prediction appears [36].

Figure 6 explains how the process of creating scenarios could be divided and used. Firstly, it depends on the timeframe that should or needs to be used. When creating a short-term scenario, usually it means that the tactical decision is made and should be accepted within 12–24 months. Secondly, there is a two to five year plan that is called a strategic planning and includes different approaches to a further development. Third is the vision, which is a rather long-term approach. The last one is a scenario that looks into the

future within ten and more years. The potential futures consist of probable and preferable variables; all of them are plausible and also possible. They evolve from certain situations and develop into uncertain futures that could be influenced from both inside and outside.

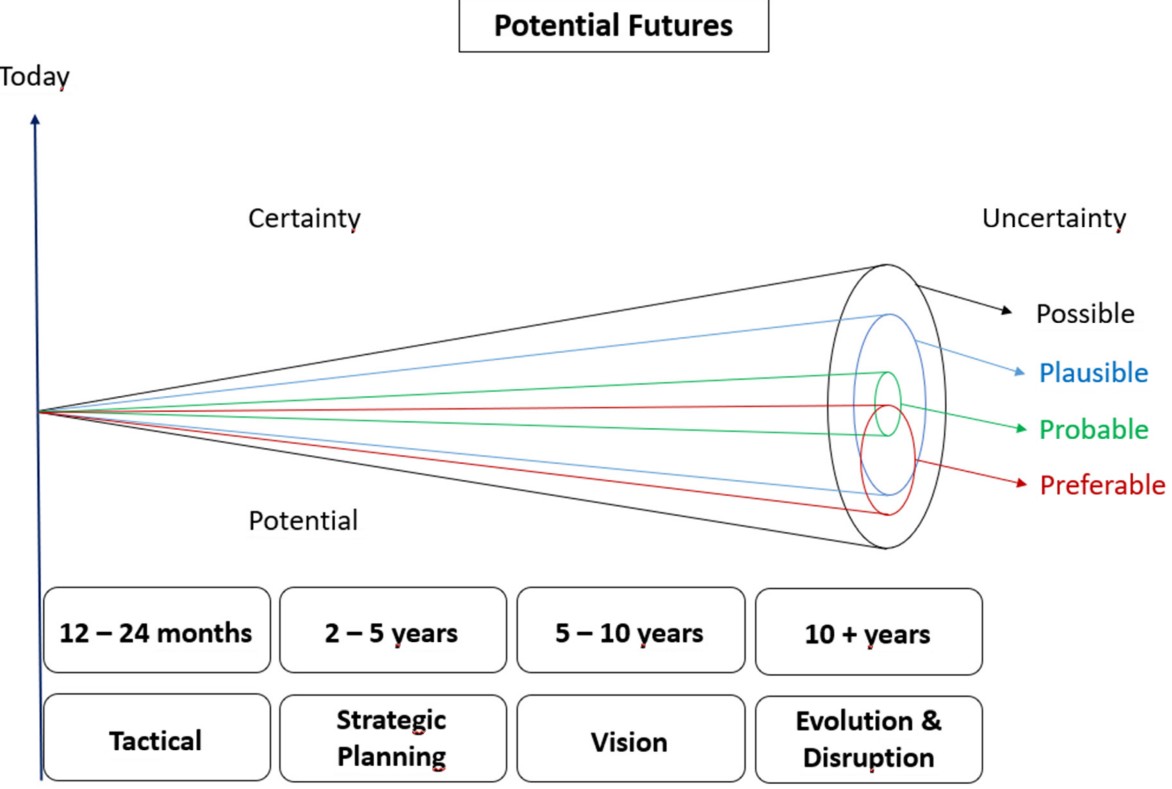

**Figure 6.** How to create scenarios. Source: Quantumrun Foresight, 2020, [37]; based on the source created by authors.

In our case, we work with the submitted economic indicators and reliable data. When designing the possible scenarios for Santorini island, we talk rather about tactical planning, which means seasons 2021 and 2022. The scenarios are based on the data and theoretical knowledge obtained from the literature; however, usually these proposed scenarios are not quantified, since their methodology is not quantitative but rather qualitative [36,37].

When looking at the development of 2020, statistical data show clear evidence of a large decrease in the number of incoming tourists to Santorini island. At the same time, the length and volume of the tourist season in 2020 had a strong impact on the local businesses. In the following text, we will present three possible scenarios for the year 2021 and 2022:

- Scenario involving the current situation, restriction and ban on traveling
- Optimistic scenario—vaccination and recovery of tourism
- Pessimistic scenario—fatal impact of 2020 season on locals and their businesses

The current situation scenario is based on the current situation and the rather stagnating number of new COVID-19 positive cases worldwide. When the number of cases was increasing, the world leaders and governments from all around Europe issued many restrictions and bans on traveling as well as closing down hotels, restaurants and other tourism-related services. Therefore, the scenario expects a similar statistical outcome for season 2021 to the year 2020, which means a rapid decline comparing with 2019 and previous years. In terms of economic impacts, local people will need to think about alternative sources of income, since the tourism industry will remain an affected sector. Thus, the social impact will be posed on families, that might disintegrate and individuals will leave to different places to find another job. In the case of a long-term tourism decline, people will not see a clear future in tourism anymore and will have to find an alternative way of living that might include moving whole families out of the island (mainly to the mainland and

Athens or other large cities). When talking about the environmental impact, this scenario brings Santorini closer to over-tourism and environmental degradation caused by stress of water.

The second scenario is an optimistic one and works with the new vaccine and its increasing coverage. Based on how many people will obtain access to the vaccine, the population will reach a collective immunity, and traveling will take place in the people's lives again. Since the vaccine has been spreading from December 2020, and the expectations are that by mid-2021 it will be accessible to most locals among European countries, the scenario expects the return of tourism on Santorini and again increasing numbers of tourists (comparing with 2020 season), slowly coming back to the numbers from previous years. Speaking of economic impacts, the scenario might be seen as an optimistic one, since people can continue running their businesses and the capacity will again reach 100% occupancy. Thus, the situation for local businesspeople will be both economically and socially beneficial. From the environmental perspective, as written in the literature review, although enough tourists will bring economic prosperity, the environmental degradation, lack of water, insufficient coverage of stable electricity, and insufficient waste management will be burning problems that will have to be addressed by local authorities.

The third, rather pessimistic scenario indicates crucial changes in tourism, new trends, and behavioral changes among people compared to travel in previous years. Additionally, the financial impacts on locals, lack of saved money, and bankruptcy of many key players will cause the majority of people depending on tourism to have to change their jobs and focus on finding anything else for their livelihood. This means that young people will leave Santorini in order to find another job elsewhere, and other people will return to an agricultural way of living, as the locals experienced before the tourism appeared. In this scenario, the economic impact will be high, and it will also significantly influence the country's GDP and local budgets. Social impact includes people changing their livelihood and finding another way of living. This might signify that families will fall apart. On the other hand, we can state that this scenario could be environmentally friendly.

### 5.3. Current Development

Regarding the season 2021, starting in June, the country will be officially open again for tourists. The prime minister Kyriakos Mitsotakis stated that by that time, Greek islands should be COVID-free. The Greek government allows international tourists with proof of vaccination, proof of recovery, or a negative COVID-19 PCR test to come. For now, the situation is rather unpredictable since many companies are crippled with debt and there has been slow vaccine rollout worldwide. Additionally, it is not clear whether people will be keen to travel in equal numbers as in previous years [38]. Based on the uncertainties that the current situation brings, any of the proposed scenarios might apply. The current situation is close to the second, rather optimistic scenario at the time of writing this article, when businesses are trying to open up for tourists.

### 5.4. Limiting Issues for This Research

One of the most significant limitations of this paper is lack of regional data for Santorini island itself. The importance of the island from the macroeconomic perspective of the whole country is not very eminent, as the research found out. On the other hand, Santorini is used as a brand and as an attractive place whose architecture and landscape attract tourists to the whole of Greece, and the touristic pressure and demand for the island are heavy. Therefore, separate regional data for tourism, employment rate in various sectors, GDP, and other economic and social variables for Santorini island would be appropriate. Regarding future research, focusing on the microanalysis of the regional data and their comparison would be adequate.

Another limitation is the speed of changes, laws, and conditions issued by governments when talking about tourism, opening borders, and setting up exact rules for travel. The COVID-19 pandemic has not finished yet, although the vaccination and immunity

after undergoing the disease contributed to lifting lockdown restrictions. Last year was full of different regulations, and with the dynamics of changes, prediction of the future is very difficult.

## 6. Conclusions

The article analyzed the impact of the COVID-19 pandemic on Greece and Santorini island in 2020. According to the statistical data, both the country and the island experienced a significant decline in the number of incoming passengers, which resulted in the drop of GDP that is highly dependent on tourism sector (20% of Greek GDP comes from tourism). With regard to Santorini and when compared to 2019, the year 2020 brought about a 65% decline in overall tourism, which is crucial for the locals in terms of employment and income. An interesting change was recorded in distribution of travel and tourism to GDP in Greece by domestic and international tourist spending. Unlike in the previous mentioned years (2015–2019), the figures show that the share of domestic spending on GDP changed and in 2020 was higher than from international tourists. Figures also prove the link between total employment rate in tourism, numbers of tourists, and increase in GDP.

The paper also proposes three possible scenarios for tourism development in the near future, which means the year 2021. The first scenario counts for the current situation and the increasing number of new COVID-19 positive cases worldwide. Related to that, there have been many restrictions and bans on international travel and therefore the scenario expects a similar statistical outcome for season 2021 to the year 2020. The second scenario is an optimistic one. Due to the new vaccination and its coverage, the scenario expects the return of tourism on Santorini and increasing numbers of tourists (comparing with 2020 season) and slowly coming back to the numbers from previous years. The third and rather negative or pessimistic scenario describes the changes in tourism, different trends, and behavioral changes among people that tend to travel. Additionally, the financial impacts on locals, lack of saved money in 2020, and bankruptcy of many key players will cause the majority of people depending on tourism to have to change their jobs and find something else for their livelihood.

It needs to be stressed that the restrictions and rules for incoming tourists to Greece are changing rapidly, but for the time being, when writing the article, the Greek authorities issued such rules that could be easily fulfilled. The willingness of international travelers to travel to Greece has been difficult to predict and depends on many determinants, such as the vaccination, cost of PCR tests, and time-limited recognition of undergoing the disease but also on transport connection and travel agencies or individual offers.

**Author Contributions:** Conceptualization, N.M., L.M. and J.H.; methodology, N.M., L.M. and J.H.; formal analysis, N.M., J.H. and L.M.; writing—original draft preparation, N.M. and L.M.; writing—review and editing, L.M and J.H. All authors have read and agreed to the published version of the manuscript.

**Funding:** This research received no external funding.

**Institutional Review Board Statement:** Not applicable.

**Informed Consent Statement:** Not applicable.

**Data Availability Statement:** MDPI Research Data Policies.

**Conflicts of Interest:** The authors declare no conflict of interest.

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
