# Peer review of "The Impact of COVID-19 on Hospitality Industry in Greece and Its Treasured Santorini Island"

_sustainability, doi:10.3390/su13147906_

Round 1

Reviewer 1 Report

Thank you for the revising. However, as an academic paper, there are still many parts that need to be revised. On the other hand, I feel that it is necessary to release the information rapidly because of the content of COVID-19 as well. Could you respond and revise at least the following aspects?

First of all, in terms of the proposed scenarios in this paper, I have the impression that they could be made by someone if they knew the rough history of this case.

So, since the authors are analyzing the local situation with the COVID-19 disaster, could you tie those figures and the status quo, optimistic, and pessimistic scenario?

Also, as a theoretical contribution in the Discussion section, could you discuss theoretical implications on the existing literature you mentioned in this paper in terms of the scenarios you proposed?

The following is a minor comment tied to the line number.

-----------------

Abstract: Please also indicate what novel results/proposals you have in this study.
20: Greece, <- Isn't this comma superfluous?
24: .[2] <- The format of the citation reference at the end of the sentence is not consistent, so please make it consistent.
36: [1;4] <- The format of multiple citation references is not consistent.
49: In a typical academic paper, you should first clearly state the problem that this research will address, and then add an explanation of the approach you will take to solve the problem. Please describe the problem you want to solve with the analysis you will use.
51: It is common to explain the structure of the paper in a separate paragraph. 
71: et al <- et al. * The same applies if it appears elsewhere.
135: 8 000 <- You may want to use a comma as a numeric separator (also see the journal's guidelines).
217: Since the scenario creation part is important for this paper, could you please explain the methodology in more detail?
234: See the journal's guideline for indentation usage.
332: Extra period
384: In a general academic paper, you need to clearly state the limiting issues for this research. Also, please explain the future work of your research based on such a limitation.

-----------------

Author Response

Dear reviewer,

I have addressed all your comments in the Cover letter that is in the attached file. 

Thank you, for your consideration.

Reviewer 2 Report

Dear authors,

This research paper describes the actual topic – The impact of COVID-19 on hospitality industry in Greece and its treasured Santorini island. Thus, authors look into the dynamic of the recent upheaval in the tourism and hospitality sector due to the COVID-19 epidemic in Greece and Santorini island. Authors compare the available statistical data showing the change in variables in the previous years with 2020 and look into the new challenges and opportunities posed by the drop in the numbers of visitors. As well, authors focus mainly on the  economic and social impact on the destination and possible future scenarios for further development in the area.

And I would like to share with authors some doubts and remarks too: it seems important to notice that  the abstract needs to be corrected - the results of the study are missing here now. As well, it would be needed to concentrate on the discussion and conclusions of the study. Thus, when developing seperate sections of "Literature review", "Discussion" and "Conclusions" it would be needed to include to the debate more newest theoretical implications, thus accessing deeper discussion and concluding insights.

Author Response

Dear reviewer,

thank you for your comments. We have addressed all of them, you will see the responses too in the attached cover letter.

Thank you again for your consideration.

With kind regards,

NM

Round 2

Reviewer 1 Report

Thank you for your response. 
Additional comments are as follows.

----------------

Abstract: In this study, the result is the presentation of the scenarios based on the data as well. Could you explain what is the uniqueness of the forecasting in Abstract?

48: Usually, the purpose of the research and the research question are discussed in the Introduction. Please clarify the purpose of the research and the research question, and then explain the structure of the analysis that will be done here to solve the research gap related to extant literature. 

176: The comma in the numerical notation has been dropped.

206: ``To analyze ...'' is not a research question, but a research objective. 

247(Figure 3): Exclude scale marks that are not needed: for example, 120%.

258: Make the indentation consistent.

284: When was the situation described in the sentence here?

347: bpoth <- Is it a typo?

381: ``Businessperson'' is adequate in terms of political correctness. 

407: ``One of the most significant limitations of this paper is lack of data for Santorini island itself.'' <- This expression reads as if the authors are not using Santorini data at all, which is quite problematic for this paper. I believe that aviation-related data, macroscopic data, literature, etc. are used, so it would be good to emphasize the lack of microscopic data available in Santorini island. 

----------------

Author Response

Dear reviewer,

please find the cover letter in the attached file. We addressed all your comments that you have mentioned in your review.

NM

This manuscript is a resubmission of an earlier submission. The following is a list of the peer review reports and author responses from that submission.

Round 1

Reviewer 1 Report

Comments

  1. The research approach seems to the lack of in depth research data and of statistics regarding substantial documentation and comparison with previous global sanitary and health threatening crises in order to extract conclusive conclusions on tourism and sustainability. It could be helpful for the scientific credibility of the paper, to critically review the literature on the impact of previous epidemic/pandemics on global tourism and compares these events to other types of global crises with final research on how Santorin’s tourism development reacted / to them comparing to Covid 19. Needless to remind that between 2000 and 2015, major tourism disruptive events occurred (not only health related) including the September 11 terrorist attacks (2001), the severe acute respiratory syndrome (SARS) outbreak (2003), the global economic crisis unfolding in 2008/2009, and the 2015 Middle East Respiratory Syndrome (MERS) outbreak. According to research none of them led to a longer-term decline in the global development of tourism, and some of them are not even notable  with only SARS (-0.4%) and the global economic crisis (-4.0%) leading to declines in international arrivals (World Bank 2020a, 2020b). This would suggest that tourism as a system had been until recently resilient to external crises shocks.(Gössling, Scott, Hall 2020) Pandemics, tourism and global change: a rapid assessment of COVID-19
  • There is a number of subjective conclusions with limited research presented to back up the claims
  • Untruthful historic facts used and basis for proposed arguments (the socioeconomic status of Santorini island in the past )  
  • Luck of statistic and references?
  • not clear use of language need to correct and specify
  • Many assumptions without data to document statements (rather hypothesis).
  • many arbitrary conclusions conjectures, assumptions statements
  • jumping to conclusion needs to compare and justify conclusion with scientific methods and evidence
  • (Not clear how this hypothesis facilitates the purpose of the paper analysis?
  • More in depth analysis is needed to justify contribution to new knowledge and not just simply applying existing knowledge

Reviewer 2 Report

This study takes up the case of Santorini, Greece, which is affected by Covid-19, and describes the efforts made so far in the tourism and the current status of the impact of Covid-19. It also discusses the possible future scenarios for this tourism industry in terms of economic, social and environmental impacts from three cases: optimistic, pessimistic and status quo.

The global impact of Covid-19 on the tourism industry is enormous, and this study may provide valuable case studies to consider how to respond more effectively in the future.

However, when viewed as an academic paper, the following major problems can be found scattered throughout the paper.

First, in the introduction, the issues of the existing research and the purpose/aim of this research are not clearly explained. Therefore, it is not possible to determine what kind of novel contribution can be obtained from this study in terms of research design.

In the literature review, there looks no systematic literature review of academic research in particular. It is questionable whether a comprehensive literature review has been conducted. It is also not clear what the issues of existing research are.

In reports of academic research, it is common to explain what methodologies were used to collect and analyze the data, but there is no such explanation.

In the result section, it is unclear how the results of this study were obtained. This is because there is no clear explanation of what analytical method was used in this paper.

As a discussion, three scenarios based on the results of this study are discussed. But it cannot be determined whether the discussion is based on the novel/original results from this study.

Reviewer 3 Report

Dear authors,

This original research paper describes the actual topic - the impact of COVID-19 on hospitality industry in Santorini, Greece. Thus, authors notice, that this paper looks into the dynamic of recent upheaval in the tourism and hospitality sector due to COVID-19 epidemic. Authors  compare the numbers of incoming visitors to the island in the previous years with 2020 and look into the new challenges and opportunities posed by the drop in the numbers of visitors. As well, authors point out, that they focus mainly on the economic, social and environmental impact on the island and possible future scenarios for further development in the area.

And I would like to share with authors some doubts and remarks too:

  1. The abstract should contain concise information on the purpose, methods used and results. In this form, it seems too general.
  2. Introduction seems quite clear, but the clarity of research question is needed to be specified.
  3. The description and presentation of methods used for the analysis is missing. Thus, methods should be explained and described in detail and clearly.
  4. The section of "Discussion" makes some conceptual doubts, as the idea of optimistic and pessimistic scenario, when living in the times of global climate change challenges, thus discussing sustainability, seems too general and needs stronger arguments.

Thus, developing the sections of "Discussion" and "Conclusions" it would be needed to include to the debate more theoretical implications, thus accessing deeper discussion and concluding insights.

Reviewer 4 Report

The authors of the manuscript presented the impact of COVID-19 on hospitality industry in Santorini (Greece). In the manuscript the authors presented an important point, although the manuscript has some drawbacks.

Main remarks:

1. Intruduction - it's too short. Please write more about hospitality, tourism (tourism development). Suggested publications:

  • Gu, X.; Hunt, C.A.; Lengieza, M.L.; Niu, L.; Wu, H.; Wang, Y.; Jia, X. Evaluating Residents’ Perceptions of Nature-Based Tourism with a Factor-Cluster Approach. Sustainability 2021, 13, 199. https://doi.org/10.3390/su13010199
  • Roman, M.; Roman, M.; Niedziółka, A. Spatial Diversity of Tourism in the Countries of the European Union. Sustainability 2020, 12, 2713. https://doi.org/10.3390/su12072713

2. Results - the manuscript has been sent to Sustainability. Please refer to sustainable development in tourism, hospitality. This is missing.

3. Conclusions - very short. What about COVID and tourism, hospitality? Suggested publications:

  • Kitamura, Y.; Karkour, S.; Ichisugi, Y.; Itsubo, N. Evaluation of the Economic, Environmental, and Social Impacts of the COVID-19 Pandemic on the Japanese Tourism Industry. Sustainability 2020, 12, 10302. https://doi.org/10.3390/su122410302
  • Roman, M.; Niedziółka, A.; Krasnodębski, A. Respondents’ Involvement in Tourist Activities at the Time of the COVID-19 Pandemic. Sustainability 2020, 12, 9610. https://doi.org/10.3390/su12229610
  • Sung, Y.-A; Kim, K.-W.; Kwon, H.-J. Big Data Analysis of Korean Travelers’ Behavior in the Post-COVID-19 Era. Sustainability 2021, 13, 310. https://doi.org/10.3390/su13010310

In your conclusions, please also answer the following questions:
• What are the directions for the future?
• What are the research gaps?
• What's new in this manuscript?

There is no critical discussion. The manuscript consists of 11 pages, of which 9 pages are the content of the manuscript (without the reference list).

4. References - the literature list includes only 21 publications. It's too short. DOI is missing (See editorial requirements).